# Regulation of Cortico-Thalamic JNK1/2 and ERK1/2 MAPKs and Apoptosis-Related Signaling Pathways in PDYN Gene-Deficient Mice Following Acute and Chronic Mild Stress

**DOI:** 10.3390/ijms24032303

**Published:** 2023-01-24

**Authors:** Fernando Yáñez-Gómez, Alfredo Ramos-Miguel, Jesús A. García-Sevilla, Jorge Manzanares, Teresa Femenía

**Affiliations:** 1Laboratorio de Neurofarmacología, IUNICS, Universitat de les Illes Balears, Crta. Valldemossa km 7.5, 07122 Palma de Mallorca, Spain; 2Health Research Institute of the Balearic Islands (IdISBa), 07120 Palma de Mallorca, Spain; 3Department of Pharmacology, University of the Basque Country (UPV/EHU), Centro de Investigación Biomédica en Red de Salud Mental (CIBERSAM), Barrio Sarriena S/N, 48940 Leioa, Spain; 4BioCruces Bizkaia Health Research Institute, Plaza de Cruces 12, 48903 Barakaldo, Spain; 5Instituto de Neurociencias de Alicante, Universidad Miguel Hernández-CSIC, Avda. de Ramón y Cajal s/n, San Juan de Alicante, 03550 Alicante, Spain; 6Redes de Investigación Cooperativa Orientada a Resultados en Salud (RICORS), Red de Investigación en Atención Primaria de Adicciones (RIAPAd), Instituto de Salud Carlos III, MICINN and FEDER, 28029 Madrid, Spain; 7Instituto de Investigación Sanitaria y Biomédica de Alicante (ISABIAL), 03010 Alicante, Spain

**Keywords:** prodynorphin, JNK, ERK, chronic mild stress, apoptosis, phosphorylation

## Abstract

The crosstalk between the opioidergic system and mitogen-activated protein kinases (MAPKs) has a critical role in mediating stress-induced behaviors related to the pathophysiology of anxiety. The present study evaluated the basal status and stress-induced alterations of cortico-thalamic MAPKs and other cell fate-related signaling pathways potentially underlying the anxiogenic endophenotype of PDYN gene-deficient mice. Compared to littermates, PDYN knockout (KO) mice had lower cortical and or thalamic amounts of the phospho-activated MAPKs c-Jun N-terminal kinase (JNK1/2) and extracellular signal-regulated kinase (ERK1/2). Similarly, PDYN-KO animals displayed reduced cortico-thalamic densities of total and phosphorylated (at Ser191) species of the cell fate regulator Fas-associated protein with death domain (FADD) without alterations in the Fas receptor. Exposure to acute restraint and chronic mild stress stimuli induced the robust stimulation of JNK1/2 and ERK1/2 MAPKs, FADD, and Akt-mTOR pathways, without apparent increases in apoptotic rates. Interestingly, PDYN deficiency prevented stress-induced JNK1/2 and FADD but not ERK1/2 or Akt-mTOR hyperactivations. These findings suggest that cortico-thalamic MAPK- and FADD-dependent neuroplasticity might be altered in PDYN-KO mice. In addition, the results also indicate that the PDYN gene (and hence dynorphin release) may be required to stimulate JNK1/2 and FADD (but not ERK1/2 or Akt/mTOR) pathways under environmental stress conditions.

## 1. Introduction

Dynorphins are endogenous opioid peptides that bind with high affinity and show significant potency at both kappa (KOR) and delta (DOR) opioid receptors [1]. The distribution of dynorphin peptides and prodynorphin (PDYN) gene expression in the brain and spinal cord suggest their involvement in a large number of conditions, including stress and anxiety [2,3,4,5]. For example, dynorphins are co-released with corticosterone into the portal circulation modulating the release of ACTH from the anterior pituitary [6,7]. Additionally, PDYN-derived peptide amounts are elevated in the thalamus and pituitary following particular forms of stress stimuli, whereas they are decreased in the pituitary after forced swimming-induced immobility [8,9,10].

At least four prior studies used PDYN-deficient mice to evaluate the role of dynorphins in anxiety. The first study [11] showed that PDYN gene deletion was associated with an enhanced response to stress stimuli, supporting an anxiolytic effect of dynorphins. In agreement, a second study using animals of the same germinal line [12] showed that anxious-like behaviors following PDYN gene deletion were accompanied by the overexpression of stress-related genes (i.e., proopiomelanocortin and corticotrophin-releasing factor genes) and reduced anxiolytic efficacy of benzodiazepines, probably due to the dysregulated expression of the γ2 and β2 subunits of the GABA_A_ receptor. In marked contrast, two other studies [3,13] described an anxiolytic behavioral phenotype following PDYN gene deletion in mice.

Besides the debate on the role of dynorphins in anxiety, the adaptations occurring in stress-related molecular pathways following PDYN gene deletion are poorly understood. Among the potential molecular mediators between dynorphins and the pathophysiological response to stress and anxiety, the mitogen-activated protein kinase (MAPK) intracellular signaling pathways might be sensible candidates. These highly conserved molecules are stimulated in response to a wide variety of extracellular stimuli and control a large number of cellular processes such as growth, proliferation, differentiation, motility, cell survival/apoptosis, and cellular stress [14,15,16,17,18,19]. Each MAPK cassette consists of three sequentially phospho-activated protein kinases (generally named MAP3K, MAP2K, and MAPK, in up-to-down-stream order) that amplify and propagate the signal [16]. Although multiple MAPK pathways have been identified in mammals, few are constitutively expressed in brain cells. These include the c-Jun N-terminal kinase 1 and 2 (JNK1/2, also known as stress-activated protein kinase or SAPK) and the extracellular signaling-regulated kinase 1 and 2 (ERK1/2) cascades. Notably, the stimulation of opioid receptors with selective peptides activates both JNK1/2 and ERK1/2 [20,21,22]. Brain MAPK pathways are also robustly stimulated by acute and chronic environmental stressors [23,24,25], which in turn regulate the activity of multiple downstream targets, some implicated in cell survival and death [26]. Furthermore, the forced-swim stress-induced activation of amygdalar ERK was abrogated in KOR knockout (KO) mice [27], and JNK deficient mice displayed blunted anxiety-like phenotypes [28]. The possibility that dynorphins may mediate stress-induced JNK1/2 and/or ERK1/2 stimulation in the brain has not been tested.

On the other hand, molecules involved in apoptotic machinery are well-known mediators of stress responses in rodents [29,30]. Notably, cortical and thalamic amounts of key components of the extracellular apoptotic pathway, including the multifunctional complex Fas/FADD (Fas-associated protein with death domain), are strongly regulated by both benzodiazepines (and other allosteric modulators of GABA_A_ receptors) [31,32] and multiple opioid ligands, including the κ-opioid receptor-selective agonist U-50488H [33,34,35,36]. Rather than increased apoptotic rates, the benzodiazepine- and opioid-induced regulation of brain Fas/FADD signaling may promote nonapoptotic, neuroplasticity-related activities [37,38]. Interestingly, MAPK-ERK and the extracellular apoptotic cascade are linked via the Akt-Pea15 (phosphoprotein enriched in astrocytes of 15 kDa) pathway. Akt-induced Pea15 phosphorylation at Ser116 switches its binding affinity from ERK1/2 to FADD, allowing the nuclear translocation of ERK1/2, and hindering apoptotic signals [39,40,41]. Of note, some studies also reported a critical role of Akt dysregulation in depression and anxiety, possibly mediated by the mTOR (mammalian target of rapamycin) cascade and the disruption of neurogenesis [42,43,44].

Using PDYN-KO mice and wildtype (WT) littermates, the present study tested the hypothesis that dynorphins mediate the stress-induced alteration of the MAPK-FADD-Akt pathways. We first investigated the cortico-thalamic neuroadaptations occurring in PDYN gene-deficient mice, possibly affecting (1) JNK1/2 and ERK1/2 MAPKs, (2) the Fas receptor/FADD extracellular apoptotic pathway, and (3) the multifunctional kinase Akt, and its substrates Pea15 and mTOR. Next, the cortical and thalamic regulation of these molecular pathways was evaluated in mice of both genotypes following acute restraint (ARS) or chronic mild (CMS) stress procedures.

## 2. Results

### 2.1. Effect of PDYN Gene Deletion on MAPK-JNK1/2 and ERK1/2 in Mouse Brain

Compared to WT animals, brain immunodensities of total (i.e., phosphorylated + non-phosphorylated) JNK1/2, ERK1/2, and Akt kinases were not significantly altered in the cortical and thalamic samples of PDYN-KO mice (see corresponding immunoblots in Figure 1, Figure 2 and Figure 3) and were, therefore, used as loading controls to normalize their respective phosphorylated forms. Thus, the amount of “activated kinase” in brain tissues refers to the ratio of phosphorylated to total kinase immunodensities. Overall, the separate and combined quantifications of JNK (i.e., JNK1 and JNK2) and ERK (i.e., ERK1 and ERK2) species yielded very similar outcomes (not shown). For practical reasons, the present study only reports combined (i.e., JNK1 + JNK2) kinase measures.

In cortical samples from PDYN-KO mice, the immunodensities of activated (phospho-Thr183/Tyr185) JNK1/2 were significantly lower (−37%, *p* < 0.05) than those in WT controls (Figure 1A). PDYN gene deletion was also associated with robust reductions in ERK1/2 activation (phospho-Thr202/Tyr204) in both the cerebral cortex (−30%, *p* < 0.01) and the thalamus (−22%, *p* < 0.01) (Figure 1B).

### 2.2. Effect of PDYN Gene Deletion on Fas/FADD Immunodensities in Mouse Brain

As previously reported [34], the immunodetection of the Fas receptor resulted in four immunoreactive bands corresponding to the monomeric (35 kDa), glycosylated (51 and 45 kDa), and aggregate (120 kDa) Fas species. Neither the cortical nor the thalamic immunodensities of these Fas receptor forms were significantly altered in PDYN-KO mice compared to WT controls (Figure 2A).

**Figure 2 ijms-24-02303-f002:**
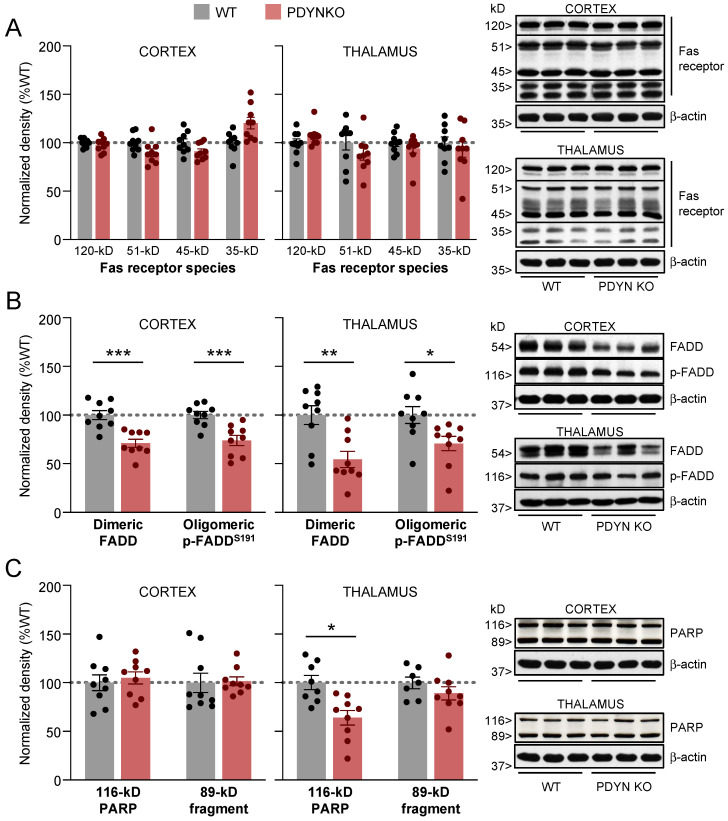
Immunodensities of (**A**) Fas receptor forms, (**B**) dimeric and oligomeric p-Ser191 FADD species, and (**C**) full-length (116-kDa) and cleaved (89-kDa) PARP in cortical and thalamic brain tissue samples from WT (grey bars; n = 9) and PDYN-KO (reddish bars; n = 9) mice. Normalized protein amounts were estimated as the ratio between each immunoreactive band to β-actin. Columns are means  ±  SEM of each experimental group and expressed in percent change from WT animals for each brain region. * *p*  <  0.05, ** *p* < 0.01, and *** *p* < 0.001, Student’s *t* test. (**A**–**C**) Representative immunoblots of cortical and thalamic Fas, FADD, and PARP species (three different animals per group) are shown on the right. The indicated molecular weights (in kDaltons, kDa) of the immunoreactive bands were estimated from in-gel-loaded, prestained protein standards.

Similar to the Fas receptor, the immunodetection of the FADD adaptor protein in mammalian brains yielded two major oligomeric species: (1) non-phosphorylated homodimers of about 51 kDa (herein named FADD), associated with its pro-apoptotic function; and (2) ~116-kDa complexes of phosphorylated FADD at Ser191 (in mice) or Ser194 (in rats and humans) (referred to as p-FADD), mainly involved in nonapoptotic activities of the FADD adaptor [36,38,45]. Compared to the WT animals, the PDYN-KO mice showed significantly lower amounts of dimeric FADD in the cortex (−29%, *p* < 0.001) and thalamus (−46%, *p* < 0.01) (Figure 2B). Similar reductions in cortical (−26%, *p* < 0.001) and thalamic (−29%, *p* < 0.05) p-FADD oligomers were observed in brain samples from the PDYN-KO mice (Figure 2B).

To address the possibility that alterations of FADD amounts in PDYN-KO mouse brains could dysregulate effector apoptotic mechanisms, the fragmentation of poly (ADP-ribose)-polymerase-1 (PARP), a major substrate in caspase-dependent and independent apoptotic pathways, was evaluated. Following apoptotic stimuli, the ~116-kDa DNA-repairing enzyme PARP is cleaved into two primary fragments of ~89 and ~31 kDa, respectively, which can be used as an indirect measure of apoptotic events in the brain [36].

In PDYN-KO mice, cortical immunodensities of the 116- and 89-kDa PARP species did not differ significantly from those in WT animals (Figure 2C). Full-length 116-kDa PARP was significantly reduced in thalamic samples from PDYN-KO animals (−36%, *p* < 0.05). The reduced density of full-length PARP was unlikely caused by greater apoptosis-related proteolytic cleavage of PARP, as the thalamic immunodensities of the 89 kDa PARP fragment were similar in both murine strains (Figure 2C). These observations indicate that PDYN gene deletion is not associated with abnormally increased cell death rates, at least in the adult mouse brain.

### 2.3. Effect of PDYN Gene Deletion on the Akt/Pea15/mTOR Pathway

The immunodensity of activated (i.e., p-Ser473) Akt was also similar in the PDYN-KO and WT mouse brains (Figure 3A), although a non-significant reduction trend was observed in the PDYN-KO thalamic samples. A similar trend was observed for Pea15 phosphorylation at Ser116 (a key cytosolic substrate of Akt) in cortical and thalamic samples from PDYN gene-deficient mice, compared to the WT controls. Of note, the brain amounts of total Pea15 were unaltered following PDYN gene deletion (see immunoblots in Figure 4). Under our experimental conditions, basal levels of mTOR phosphorylation at Ser2448 (another relevant downstream substrate of the Akt signaling pathway) were barely detectable in brain samples of both murine strains (see corresponding immunoblots in Figure 4). Therefore, it was not possible to provide a quantitative estimate comparing brain p-mTOR levels in PDYN-KO and WT stress-naïve mice.

**Figure 3 ijms-24-02303-f003:**
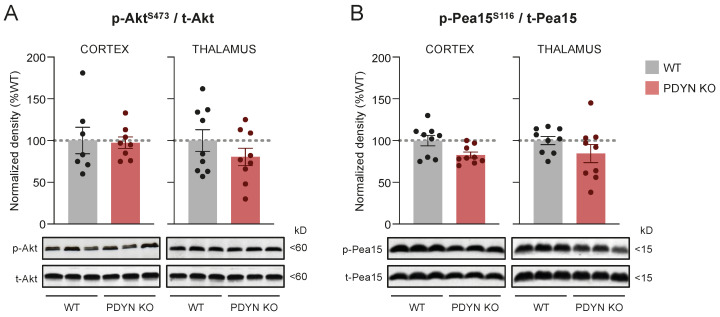
Immunodensities of activated (**A**) Akt and (**B**) Pea15 in cortical and thalamic brain tissue samples from WT (grey bars; n = 9) and PDYN-KO (reddish bars; n = 9) mice. Normalized protein activation was estimated as the ratio between phosphorylated (i.e., p-Ser473 Akt or p-Ser116 Pea15) to non-phosphorylated protein species. Columns are means  ±  SEM of each experimental group and expressed in percent change from WT animals for each brain region. (**A**,**B**) Representative immunoblots of cortical and thalamic Akt and Pea15 phosphorylated and non-phosphorylated species (three different animals per group) are shown at the bottom. The indicated molecular weights (in kDaltons, kDa) of the immunoreactive bands were estimated from in-gel-loaded, prestained protein standards.

### 2.4. Effects of Acute and Chronic Stress on Select Signaling Pathways in Cortical and Thalamic Samples from WT and PDYN-KO Mice

The potential role of the PDYN gene in mediating the stress-induced dysregulation of the studied intracellular signaling pathways was tested by comparing the effects of acute restraint (ARS) or chronic mild (CMS) stress procedures versus basal stress conditions (i.e., undisturbed mice) on cortical and thalamic densities of activated MAPKs (JNK/ERK), select proteins of the apoptotic pathway (FADD/PARP), and the phosphorylation of Akt/Pea15/mTOR signaling proteins, in both PDYN-KO mice and WT littermates [2,15]. All neurochemical datasets obtained in the subsequent immunoblotting assays in brain samples from undisturbed, and stressed WT and PDYN-KO animals were analyzed by two-way analysis of variance (TW-ANOVA), and the main result outputs were summarized in Table 1. Note that the degrees of freedom did not match up in all the analyses as group sizes varied due to the following reasons: (1) outliers were excluded from the analyses whenever detected by Grubb’s test, or (2) some samples were finished before completing the neurochemical analyses.

#### 2.4.1. ARS and CMS Effects on Cortical Targets in WT and PDYN-KO Mice

As expected, the phospho-activation of cortical JNK1/2 (6.6–7.1-fold increase, *p* < 0.001) and ERK1/2 (3.2–3.5-fold increase, *p* < 0.001) was dramatically enhanced in WT animals after both the ARS and CMS procedures, compared to undisturbed mice (Figure 4A,B). Interestingly, PDYN gene deletion had a different influence on cortical JNK1/2 and ERK1/2 stimulations following stress stimuli. Thus, ARS- and CMS-mediated JNK1/2 activations observed in WT animals were largely prevented in PDYN-KO mice (response inhibition: 67–70%, *p* < 0.001) (Figure 4A). In marked contrast, PDYN-KO animals displayed comparable p-ERK1/2 cortical levels than those in WT littermates following both ARS and CMS procedures (response inhibition: 6–8%, *p* > 0.05) (Figure 4B).

CMS (3.8-fold, *p* < 0.001), but not ARS (1.6-fold, *p* > 0.05), markedly upregulated dimeric FADD in the cerebral cortex of WT mice (Figure 4C). Remarkably, the CMS-induced upregulation of dimeric FADD in the PDYN-KO mouse cortex (2.4-fold increase, *p* < 0.001) was significantly smaller than that in the WT mice (response inhibition: 49%, *p* < 0.01) (Figure 4C). Both ARS (3.5-fold, *p* < 0.001) and CMS (3.6-fold, *p* < 0.001) procedures dramatically increased the cortical density of oligomeric p-FADD in the WT animals, and similar upregulations were observed in the PDYN-KO mice after the same stress stimuli (3.5–4.4-fold increases, *p* < 0.001), as compared to their respective non-stressed controls (Figure 4D). These data suggest that the stress-induced upregulation of dimeric FADD, but not its phosphorylation, rely (at least in part) on PDYN gene expression. Despite the massive upregulation of cortical FADD species, PARP fragmentation remained unchanged across the groups (Figure 4E), indicating that stress-induced FADD does not necessarily involve an increase in apoptotic rates.

**Figure 4 ijms-24-02303-f004:**
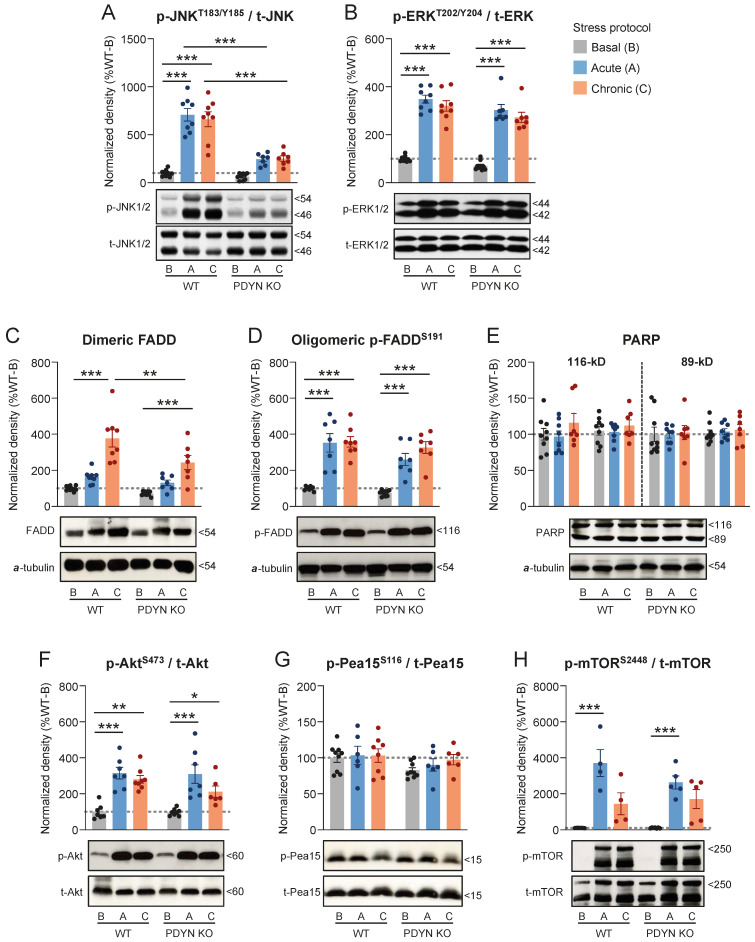
Cortical immunodensities of (**A**) activated p-Thr183/Tyr185 JNK1/2, (**B**) activated p-Thr202/Tyr204 ERK1/2, (**C**) dimeric FADD, (**D**) oligomeric p-Ser191 FADD, (**E**) full-length and cleaved PARP, (**F**) activated p-Ser473 Akt, (**G**) p-Ser116 Pea15, and (**H**) activated p-Ser2448 mTOR in WT and PDYN-KO mice exposed to acute restraint (**A**, blue bars) or chronic mild (**C**, orange bars) stress procedures, as compared to basal (**B**, grey bars) stress levels in undisturbed animals. Normalized protein amounts were estimated as the ratio between the corresponding immunoreactive band to total enzyme or α-tubulin. Columns are means  ±  SEM of n = 7–9 mice per experimental group and expressed in percent change from wildtype-basal (WT-B) mice. All datasets were analyzed by TW-ANOVA (see Table 1). * *p*  <  0.05, ** *p*  <  0.01, and *** *p*  <  0.001, TW-ANOVA followed by Tukey’s post hoc test. (**A**–**H**) Representative immunoblots of phosphorylated and/or non-phosphorylated species of the indicated proteins are shown at the bottom. The indicated molecular weights (in kDaltons, kDa) of the immunoreactive bands were estimated from in-gel-loaded, prestained protein standards.

Furthermore, Akt phosphorylation at Ser473 was similarly upregulated in cortical samples of WT (2.8–3.2-fold increase, *p* < 0.001) and PDYN-KO (2.9–4.1-fold increase, *p* < 0.001) following ARS and CMS exposure (Figure 4F), indicating that the PDYN genotype had no impact on the stress-mediated stimulation of cortical Akt. Surprisingly, Akt hyperactivation did not significantly influence Pea15 phosphorylation at Ser116, which remained unaltered after the stress procedures in the same cortical samples (Figure 4G). On the other hand, the cortical immunodensities of p-Ser2448 mTOR were similarly increased in both murine strains in response to ARS (263-370-fold increase, *p* < 0.001) and, to a lesser extent, CMS (144–171-fold increase, *p* > 0.05) (Figure 4H). Of note, cortical amounts of p-mTOR in mice with basal stress levels were probably below the linear range of detection, and the data provided should be taken qualitatively rather than quantitatively.

#### 2.4.2. ARS and CMS Effects on Thalamic Targets in WT and PDYN-KO Mice

In marked contrast, stress exposure did not significantly modify JNK1/2 activation in the thalamus of WT and PDYN-KO mice compared to their respective basal groups (Figure 5A). Unfortunately, ERK1/2 was not addressed in these samples due to limitations in tissue availability.

In thalamic samples from the WT mice, significant upregulations of dimeric FADD were quantified in ARS (+78%, *p* < 0.001)- and CMS (+109%, *p* < 0.001)-exposed mice, compared to undisturbed animals with basal stress levels (Figure 5B). Interestingly, the stress-induced upregulations of dimeric FADD reported in the WT animals were not observed in the ARS- or CMS-exposed PDYN-KO mice (response inhibition: 80–88%, *p* < 0.001) (Figure 5B). In marked contrast, the thalamic densities of oligomeric p-FADD remained unchanged in the WT mice exposed to stress stimuli, and only CMS-exposed PDYN-KO animals had significantly lower p-FADD densities, as compared to WT littermates exposed to the same stress procedure (−44%, *p* < 0.05) (Figure 5C). However, this difference in thalamic p-FADD levels may be attributed exclusively to genotype but not the genotype×stress interaction effects (see Table 1).

Unlike the observations in the cerebral cortex, the thalamic samples of the ARS- and CMS-exposed animals displayed similar amounts of activated Akt to those in the basal group, regardless of the genotype (Figure 5D). Accordingly, the thalamic immunodensities of p-Ser116 Pea15 did not differ significantly across the experimental groups (Figure 5E).

## 3. Discussion

The present study aimed to unmask some molecular mechanisms that may contribute to the anxiogenic endophenotype in mice lacking the PDYN gene. The results showed that, under basal conditions, PDYN-KO animals display lower brain amounts of activated JNK1/2 and ERK1/2 MAPKs and FADD protein species, which are critical components of signaling pathways controlling crucial cellular functions related to neuroplasticity and cell fate. Interestingly, the present findings further suggested that the cortical hyperactivation of JNK1/2 and cortico-thalamic FADD upregulation following stressful stimuli may be dynorphin-dependent mechanisms, as these effects were not observed in stressed PDYN-KO mice. As the hyperactivation of ERK1/2 and Akt/mTOR pathways in the same cortical samples of ARS- and CMS-exposed animals was not prevented by PDYN gene deletion, the control of dynorphins over the JNK1/2 and FADD cascades might be a selective mechanism. These ideas are graphically summarized in Figure 6.

Although the role of the PDYN gene in anxiety-like behaviors remains controversial [1], the present work used a PDYN-KO murine strain displaying a greater vulnerability to developing anxiety-like behaviors [11,12] to investigate the possible compensatory effects occurring in the cortico-thalamic, stress-related signaling pathways. PDYN-KO mice displayed reduced basal activation of the JNK1/2 and ERK1/2 MAPKs, as compared to WT littermates. Similarly, the multifunctional adaptor of the extracellular apoptotic pathway FADD, and its oligomeric form phosphorylated at Ser116, were downregulated in the same brain samples without changes in the Fas receptor species or PARP fragmentation. In terms of the signaling pathways studied, the PDYN gene deletion had a similar impact on both the cortex and the thalamus, suggesting that the reported adaptations to PDYN gene deficiency may begin early in brain development. Perhaps, the neural adaptations occurring in another PDYN-KO strain showing an anxiolytic endophenotype [3,13] evolved over the generations in opposite directions than those presented here, which could explain the contrasting behaviors against stressful stimuli of the two PDYN-KO murine strains. Whether the reported adaptations of MAPKs and FADD signaling pathways observed in PDYN-KO brains actually contribute to predisposing these mutant animals to develop an exacerbated response against environmental stress should be fully elucidated in future studies. In this context, other signaling molecules that should be investigated are those involved in oxidative stress and the production of reactive oxygen species (ROS). Thus, environmental stressors accelerate the machinery of ROS production in the brain [46,47,48], and, on the other hand, dynorphins tightly regulate brain oxidative stress mechanisms [49,50].

The present work further addressed the impact of acute and chronic stress stimuli on brain amounts of the selected molecular pathways in the presence or absence of the PDYN gene. In agreement with prior studies reporting the overstimulation of MAPKs in rodent brains following different stress protocols [25,26,27,51], the exposure of WT animals to stressful environments led to a robust and sustained overactivation of the JNK1/2, ERK1/2, FADD, and Akt signaling pathways (Figure 6). These molecular changes did not show sensitization or tolerance to repeated stress exposure, as similar results were obtained in ARS and CMS protocols. To address the possibility that the aberrant stimulation of the studied signaling pathways (especially those of FADD and JNK1/2) may activate cell death mechanisms, we evaluated PARP fragmentation, a well-known substrate of caspase-dependent and -independent apoptotic mechanisms [52]. No alterations were found in the cortical cleavage of this polymerase, suggesting that stress exposure did not significantly enhance apoptotic cell death rates in cortical tissues, despite the large upregulation in apoptosis-related molecules such as FADD and JNK1/2. Unfortunately, due to the limited tissue sample availability, we could not assess PARP (and ERK1/2) in the thalamic region. Although we cannot discard that some increase in apoptotic rates might have occurred in the thalamus of ACS- and/or CMS-exposed mice [53], it seems unlikely that the modest increase in thalamic FADD (as compared to that in the cortex) results in greater cell death rates than those in the cortical tissue of the same animals. Instead, all these signaling molecules are well-known mediators of gene regulation and neural plasticity [54,55]. Arguably, the aberrant stimulation of MAPKs, FADD, and Akt/mTOR signaling pathways may rather contribute to stress-mediated changes in gene expression profiles and the subsequent neuroadaptations underlying the anxiety-like behavioral responses (Figure 6).

Importantly, the dramatic stress-induced upregulation of two stress-related signaling molecules—JNK1/2 and FADD—observed in the brains of the WT animals did not occur in mice lacking the PDYN gene exposed to the same stress procedures. Of note, we did not assess the effect of the present stress procedures on dynorphin amounts in WT animals. However, stress-induced dynorphin release has been previously demonstrated under multiple types of environmental stressors [56]. Although remote, we should also consider that JNK1/2 and FADD stimulations may be attributed to alternative causes other than stress-induced dynorphin release and the subsequent activation of KOR/DOR. In contrast to JNK1/2 and FADD, ERK and Akt/mTOR cortical hyperactivations were similarly observed in both the WT and PDYN-KO mice, suggesting that the PDYN gene exerts control over select signaling pathways. As PDYN-KO mice are more vulnerable to environmental stressors [11,12], dynorphins and the stimulated downstream pathways may contribute to mitigating the cellular and molecular mechanisms leading to anxiety [1] (Figure 6).

The role of MAPKs in stress and anxiety has been broadly studied and reviewed [23,24,57]. In contrast to our findings, JNK1-deficient mice displayed an anxiolytic endophenotype, compared to WT littermates, which was consistent with the behavioral response observed in mice repeatedly treated with the selective JNK1/2 inhibitor DJNKI-1 [28]. However, all these JNK1-mediated behavioral effects were attributed to dysregulated hippocampal neurogenesis. Therefore, JNK1 stimulation may contribute to blocking neurogenesis in the hippocampus and favor anxiogenic behaviors, whereas, in the cerebral cortex, where the neurogenesis mechanisms are not operative, the hyperactivation of JNK1/2 may have an anxiolytic effect by tuning down other stress-related signaling pathways. Conversely, prior studies reported that forced swim and social defeat stress stimuli increased ERK1/2 phosphorylation in mouse striatum and amygdala, respectively, in a KOR-dependent manner [27,51]. In contrast, under our experimental conditions, cortical ERK1/2 hyperactivation was not prevented by PDYN gene deletion. Of note, dynorphins are endogenous ligands of both KOR and DOR, which enkephalins can also stimulate. Hence, in the present study, stress-induced ERK1/2 activation in the PDYN-KO animals might have occurred via enkephalin-mediated KOR stimulation. Brain region differences (striatum and amygdala versus cortex) may also explain these contrasting results.

Intracellularly, ERK1/2 and FADD are linked via the Akt-Pea15 pathway [40]. Indeed, Akt stimulation mirrored the upregulation of ERK1/2 and FADD under stress stimuli. Surprisingly, stress-induced Akt activation was not followed by increased Pea15 phosphorylation at Ser116. Instead of Pea15, Akt downstream targets included mTOR. In cortical and hippocampal neurons, the Akt-mTOR signaling pathway can modulate the activity of both ionotropic and metabotropic glutamate receptors, thereby becoming a key regulator of the neuroplasticity mechanisms orchestrating learning and memory [58].

In contrast, prior studies showed that stress exposure was associated with downregulated Akt-mTOR signaling in the mouse brain, and antidepressant drugs reverted those effects [59,60]. Many factors may account for these discordant observations across the studies, including the stress protocol applied and/or the brain areas studied. The microinjection of selective JNK1/2, ERK1/2, and Akt inhibitors into specific brain areas will be needed in future studies to address the neuroanatomical specificity of these controversial data.

In conclusion, PDYN gene deletion is associated with reduced activation of JNK1/2, ERK1/2, and FADD, suggesting that basal brain dynorphin levels may exert a tonic control over these signaling cascades. Both acute and chronic stress stimuli are followed by a concomitant activation of multiple intracellular signaling pathways, of which, cortical JNK1/2 and cortico-thalamic FADD appear to rely on the PDYN gene.

## 4. Materials and Methods

### 4.1. Animals

Male pre-prodynorphin gene double-deficient (PDYN-KO) mice and WT littermates were used in all the experiments. The generation of mice lacking the pre-prodynorphin gene was previously described [61]. For neurochemical analyses, WT and PDYN-KO mutant mice were matched (age: 2-3 months; weight: 25–35 g). The mice were maintained at a constant temperature of 23 ± 2 ºC and in a 12 h dark/light cycle (light from 8 a.m. to 8 p.m.), with free access to food and water. The Institutional Committee Board on Experimental Animal Welfare at Miguel Hernández University reviewed and approved all experimental procedures, which were designed in compliance with the European Council Directive of 24 November 1986 (86/609/EEC).

### 4.2. Acute and Chronic Stress Procedures

WT (n = 25) and PDYN-KO (n = 23) mice were randomly assigned to one of the following protocols: basal stress (n = 18), acute restraint stress (ARS; n = 15), or chronic mild stress (CMS; n = 15) (see Appendix A for detailed group sizes). Control animals (basal stress) were left undisturbed in their home cages for the entire duration of the assays and were only handled for routine housekeeping tasks. For ARS delivery, WT and PDYN-KO mice were individually confined into an acrylic cylindrical tube (inner size 10 × 3 × 3 cm), with small holes to allow normal breathing. After 30 min, the animals were released and returned to their home cages. For the CMS procedure, mice were exposed daily to restrain stress for a week in combination with other mild stressors to avoid habituation and add an element of unpredictability to the stressor. The following stressors were applied several times (2–3) a day: wet cage, food deprivation, restraint stress, period of stroboscopic illumination (150 flashes·min^−1^), inversion of light/dark cycle, tilted cage (45°), and loud noise (90–105 dB) [62].

### 4.3. Brain Tissue Preparation

Two hours and thirty minutes after the stress procedures, the mice were killed by decapitation. The brains were immediately removed, and the whole cerebral cortices and the thalamus were dissected and frozen at −80 °C until the assays [63,64,65]. Total brain tissue homogenates were obtained, as described in detail [45]. Briefly, frontoparietal cortical (170–180 mg) and thalamic (50–60 mg) samples were homogenized (1:15 *w*/*v*) in ice-cold 50 mM Tris-HCl buffer pH 6.8, containing 1 mM EDTA, 2% sodium dodecyl sulfate (SDS), and 1% of each protease (P8340, Sigma-Aldrich, Heidelberg, Germany) and phosphatase (P2850, Sigma-Aldrich) inhibitor cocktails, and then sonicated for 10 s. The protein concentration was determined using bicinchoninic acid for the colorimetric detection of the cuprous cation (BCA Protein Assay Reagent; Pierce Chemical Co., Rockford, IL, USA). Aliquots of brain samples were combined with equal volumes of loading buffer (100 mM Tris HCl, pH 6.8, 3% SDS, 20% glycerol, 5% β-mercaptoethanol, and ~0.1% bromophenol blue), denatured at 95 °C for 4 min, and stored at −80 °C until use.

### 4.4. Immunoblot Assays and Quantification of Target Proteins

Brain protein samples (40 µg/well) were separated in 10% SDS-PAGE mini gels (Bio-Rad Laboratories, Hercules, CA, USA) and transferred to nitrocellulose membranes. All the gels were also loaded with a broad range of molecular weight prestained standards (Cat#161-0317, Bio-Rad) to estimate the molecular size of the immunoreactive bands. The membranes were incubated for 1 h in phosphate-buffered saline (PBS, in experiments where the target was a non-phosphorylated protein) or Tris-buffered saline (TBS, in experiments where the target was a phosphorylated protein) containing 5% non-fat dry milk, 0.1–0.2% Tween-20, and 0.5% bovine serum albumin to block nonspecific protein-binding sites (blocking solution). After washing, the membranes were incubated overnight at 4 °C in a blocking solution containing the appropriate primary antibody. The characteristics of all primary antibodies used in the present study are provided in Table 2. Finally, the blots were incubated with the corresponding secondary antibody (anti-rabbit or anti-mouse IgG) conjugated to horseradish peroxidase (1:5000 dilution; Cell Signaling Technology, Danvers, MA, USA) in a blocking solution for 1 h at room temperature. Following incubation, in commercial ECL reagents (Amersham International, Buckinghamshire, UK), the chemiluminescent signal of bound antibody was visualized by exposure to autoradiographic film (ECL Hyperfilm, Amersham) for 1–20 min.

All membranes were stripped and reprobed with an antibody against a housekeeping protein to control for possible inaccuracies at sample loading. Thus, phosphorylated species of JNK1/2, ERK1/2, Akt, Pea15, and mTOR were corrected by the amounts of the corresponding total protein (Table 2). In experiments comparing unstressed WT versus PDYN-KO mice, Fas, FADD, and PARP immunoblots were reprobed with the anti-β-actin monoclonal antibody (Table 2). As environmental stress may alter β-actin gene expression [66], α-tubulin was used as the reference protein in experiments involving ACS- and CMS-exposed mice.

The integrated optical densities (IOD) of the specific immunoreactive bands were quantified by densitometric scanning (GS-800 Imaging Densitometer, Bio-Rad). The raw IODs in an autoradiogram were first divided (i.e., normalized) by the IODs of the corresponding housekeeping protein in the same membrane. For a direct comparison, brain samples of at least three unstressed WT (i.e., control) mice were run in all gels, together with other samples from the tested groups. The mean of normalized IODs across the control samples loaded in the same gel was set as the reference value, and the immunodensities of all samples in that gel were calculated in percent change from the reference. Each sample was assessed in 3–6 different gels, and the mean across all experimental measures was used as the final estimate.

### 4.5. Data and Statistical Analyses

Data were analyzed with JMP16 (SAS Institute, Cary, NC, USA) and plotted in GraphPad Prism™, version 9.0 (GraphPad Software, San Diego, CA, USA). The results are expressed as mean values ± SEM. A two-tailed Student‘s *t*-test was used when comparing unstressed (basal) PDYN-KO mice versus WT littermates. The effects of acute and chronic stress across genotypes were evaluated with two-way analyses of variance (TW-ANOVA) followed by the Tukey HSD post hoc test. The level of significance was set to *p* < 0.05.

**Table 2 ijms-24-02303-t002:** Antibodies used for the detection and quantification of target proteins.

Protein	Immunogen	Host	Catalogue (Clone)	Batch	Vendor
p-JNK1/2	Human JNK (peptide containing p-Thr183/Tyr185)	Rabbit	9251	10	Cell Signaling, USA
JNK1/2	Human JNK2 (GST fusion protein)	Rabbit	9252	6	Cell Signaling, USA
p-ERK1/2	Human p44 MAPK (peptide containing p-Thr202/Tyr204)	Rabbit	9101	20	Cell Signaling, USA
ERK1/2	Human p42 MAPK (synthetic peptide)	Rabbit	442704	D33075	Calbiochem, FRG
Fas receptor	Mouse Fas receptor (C-terminal residues)	Rabbit	sc-716	F1306	Santa Cruz, USA
FADD	Human FADD (28–208 residues)	Rabbit	sc-5559	K1407	Santa Cruz, USA
p-FADD	Mouse p-FADD (peptide containing p-Ser191)	Rabbit	2785	1	Cell Signaling, USA
PARP	Human PARP-1 (215–228 residues)	Rabbit	512729	KP8501	Calbiochem, FRG
p-Akt	Mouse Akt1 (peptide containing p-Ser473)	Rabbit	9271	10	Cell Signaling, USA
Akt	Human Akt1 (88-100 residues)	Rabbit	530311	D35774	Calbiochem, FRG
p-Pea15	Human PEA-15 (peptide containing p-Ser116)	Rabbit	44-836G	0103	BioSource Inc., USA
Pea15	Human PEA-15 (peptide containing Leu60)	Rabbit	2780	1	Cell Signaling, USA
p-mTOR	Human p-mTOR (p-Ser2448)	Rabbit	2971	14	Cell Signaling, USA
mTOR	Human p-mTOR	Rabbit	2972	7	Cell Signaling, USA
β-actin	Human β-actin (2-16 residues)	Mouse	A1978 (AC-15)	016K4817	Sigma-Aldrich, USA
α-tubulin	Sea urchin sarkosyl-resistant filaments	Mouse	T6074 (B-5-1-2)	6	Sigma-Aldrich, USA

## Figures and Tables

**Figure 1 ijms-24-02303-f001:**
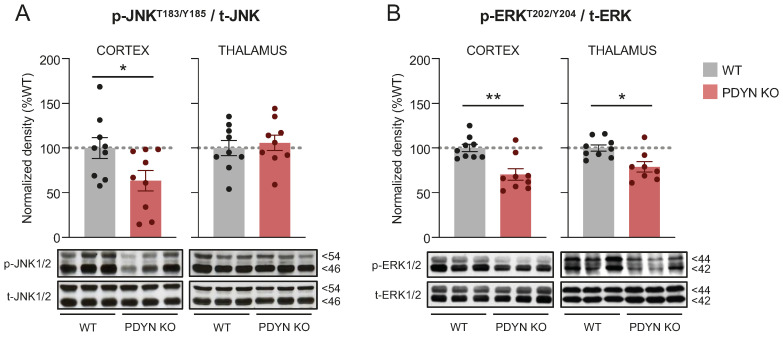
Immunodensities of activated (**A**) JNK1/2 and (**B**) ERK1/2 in cortical and thalamic brain tissue samples from WT (grey bars; n = 9) and PDYN-KO (reddish bars; n = 9) mice. Normalized enzyme activation was estimated as the ratio between phosphorylated (i.e., p-Thr183/Tyr185 JNK1/2 or p-Thr202/Tyr204 ERK1/2) to non-phosphorylated protein species. Columns are means  ±  SEM of each experimental group and expressed in percent change from WT animals for each brain region. * *p*  <  0.05, and ** *p* < 0.01, Student’s *t* test. (**A**,**B**) Representative immunoblots of cortical and thalamic JNK and ERK phosphorylated and non-phosphorylated species (three different animals per group) are shown at the bottom. The indicated molecular weights (in kDaltons, kD) of the immunoreactive bands were estimated from in-gel-loaded, prestained protein standards.

**Figure 5 ijms-24-02303-f005:**
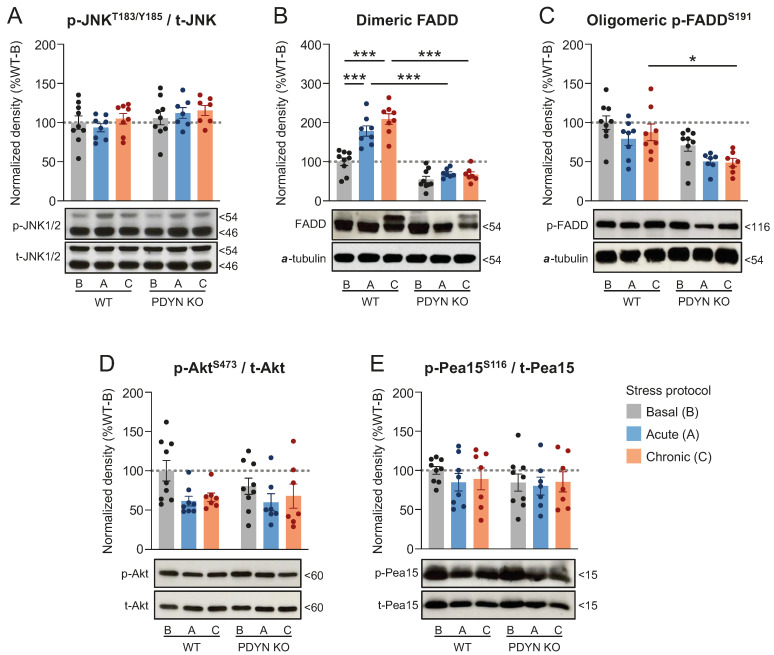
Thalamic immunodensities of (**A**) activated p-Thr183/Tyr185 JNK1/2, (**B**) dimeric FADD, (**C**) oligomeric p-Ser191 FADD, (**D**) activated p-Ser473 Akt, and (**E**) p-Ser116 Pea15 in WT and PDYN-KO mice exposed to acute restraint (A, blue bars) or chronic mild (C, orange bars) stress procedures, as compared to basal (B, grey bars) stress levels in undisturbed animals. Normalized protein amounts were estimated as the ratio between the corresponding immunoreactive band to total enzyme or α-tubulin. Columns are means  ±  SEM of n = 7–9 mice per experimental group and expressed in percent change from wildtype-basal (WT-B) mice. All datasets were analyzed by TW-ANOVA (see Table 1). * *p*  <  0.05, and *** *p*  <  0.001, TW-ANOVA followed by Tukey’s post hoc test. (**A**–**E**) Representative immunoblots of phosphorylated and/or non-phosphorylated species of the indicated proteins are shown at the bottom. The indicated molecular weights (in kDaltons, kDa) of the immunoreactive bands were estimated from in-gel-loaded, prestained protein standards.

**Figure 6 ijms-24-02303-f006:**
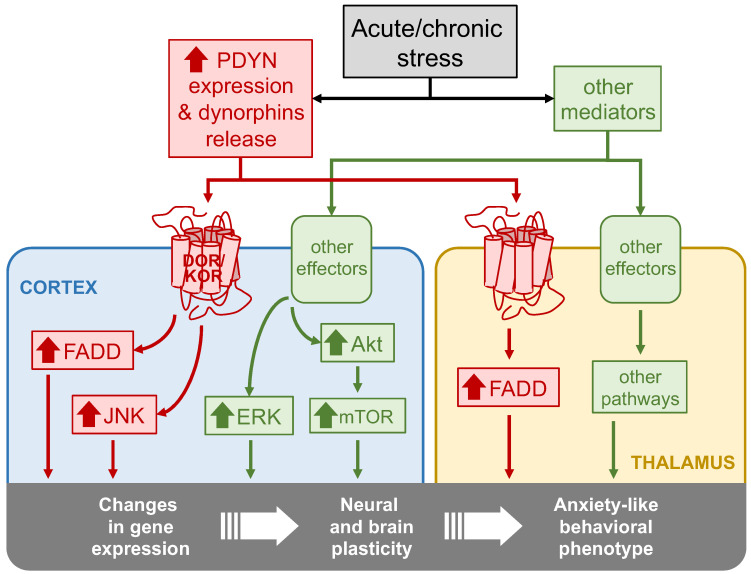
Flow diagram summarizing the major findings of the study.

**Table 1 ijms-24-02303-t001:** Results of TW-ANOVA reporting the effects of the stress procedures, PDYN gene deletion, and their interaction on cortical and thalamic immunodensities of the studied proteins.

Brain Area	Target	Whole Model			Stress Protocol		Genotype		Interaction	
		df	F-Ratio	*p*-Val	F-Ratio	*p*-val	F-Ratio	*p*-Val	F-Ratio	*p*-Val
Cortex	p-JNK1/2	5, 42	26.6	<.001	35.3	<.001	39.6	<.001	8.38	<.001
	p-ERK1/2	5, 42	29.8	<.001	139	<.001	9.62	0.003	0.12	0.820
	FADD	5, 42	20.8	<.001	42.7	<.001	9.98	0.003	2.71	0.078
	p-FADD	5, 41	22.5	<.001	53.0	<.001	47.4	0.035	0.83	0.444
	PARP 116 kDa	5, 42	1.08	0.384	0.41	0.668	3.33	0.075	0.84	0.440
	PARP 89 kDa	5, 42	0.12	0.988	0.21	0.809	0.17	0.682	0.01	0.994
	p-Akt	5, 38	13.0	<.001	30.6	<.001	1.86	0.181	0.54	0.588
	p-Pea15	5, 38	1.19	0.331	0.71	0.499	3.45	0.071	0.29	0.748
	p-mTOR	5, 27	12.7	<.001	29.6	<.001	0.65	0.427	1.42	0.261
Thalamus	p-JNK1/2	5, 42	1.20	0.324	0.38	0.686	4.68	0.036	0.15	0.861
	FADD	5, 42	40.7	<.001	20.8	<.001	140	<.001	11.9	<.001
	p-FADD	5, 42	6.42	<.001	4.00	0.026	24.6	<.001	0.23	0.796
	p-Akt	5, 41	2.11	0.084	4.34	0.020	0.53	0.471	0.58	0.566
	p-Pea15	5, 42	0.48	0.792	0.54	0.585	0.21	0.650	0.50	0.612

## Data Availability

Data may be made available to other researchers upon request.

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
