# Peer review of "Regulation of Cortico-Thalamic JNK1/2 and ERK1/2 MAPKs and Apoptosis-Related Signaling Pathways in PDYN Gene-Deficient Mice Following Acute and Chronic Mild Stress"

_ijms, 2023, doi:10.3390/ijms24032303_

Round 1

Reviewer 1 Report

Your articles provide relevant information about the cortico-thalamic JNK1/2 and ERK1/2 MAPKs and apoptosis-related signaling pathways. However, the inaccurate choice of words, the result description incorrectly, and the manuscript format are causes of the decline in the value of the information presented. So, I think this article should be revised.

Author Response

Reviewer #1:

Abstract

Comment: "Shown in introduction section”

Response: Study goals were rephrased in the Abstract to avoid the redundancy.

Comment: “Just described the significant results”

Response: Nonsignificant results were removed from the abstract.

Comment: “Confusing wording”

Response: Sentence rephrased.

Introduction

Comment: “The reference format in this paper should conform to journal standards. Like [1, 2]”

Response: References were automatically introduced with Mendeley software, using the IJMS style plugin. In the current version, references were manually edited to meet the actual journal style.

Comment: “The manuscript should show line numbers.”

Response: Line numbers are now shown.

Comment: “The front part and the back part seem to have no relationship”

Response: Sentence rephrased.

Comment: “What do you mean constitutively?”

Response: We agree with the Reviewer, “Constitutively active” is confusing. The term was replaced by “constitutively expressed”, which has a concrete meaning, and applies to brain JNK and ERK expression.

Results

Page 6

Comment: “The fold change in this section should be described respectively, just like in chapter 2.1”

Response: Comment attended all along the Results section.

Comment: “Confusing wording”

Response: The sentence was rephrased.

Comment: “Deletion”

Response: Deleted.

Page 7

Comment: “Spelling”

Response: Spelling corrected.

Comment: “The image of actin must put in figrue2A”

Response: The beta-actin blots are now shown. Since we are not reporting significant changes for that particular protein, we did not consider necessary showing the housekeeping blot, as the figure was esthetically more pleasing without those blots.

Comment: “Separate Description”

Response: The description was split.

Page 8

Comment: “P value??”

Response: Statistics are now indicated

Page 9

Comment: “Confusing wording”

Response: Sentences modified to clarify the argument.

Comment: “Is the WT sample included?”

Response: Yes, the experiment included WT animals as controls. This issue is now is specified.

Page 10

Comment: “This experiment should determine the expression levels of dynorphins after treatment with CMS, ARS, and basal in WT and PDYN-KO mice.”

Response: We did not measure circulating or brain tissue levels of dynorphins in these animals. Therefore, we switched “dynorphins” for “PDYN gene” in that particular sentence to avoid misinterpretations. Unfortunately, the samples are now exhausted, and we cannot provide evidence of increased dynorphin release following stress stimuli under our experimental conditions. We now acknowledge this important limitation of the study in the Discussion (see page 17-18), so that readers will be aware of this issue. Following that warning, we added a comment arguing that stress-induced dynorphin release is a well-supported observation across the scientific literature that different authors reported previously. Please, find a more elaborated description on this topic in Knoll & Carlezon Brain Res. 2010 (now cited).

Comment: “The sentences should be shown in the introduction section.”

Response: The second paragraph of the Introduction already described this issue, and therefore, the highlighted sentence has been deleted.

Comment: “Confusing wording”

Response: Explanation of missing data points was simplified.

Comment: “Check the grammar”

Response: Checked.

Page 11

Comment: “Poor description of results”

Response: Whenever relevant, the response inhibition is now documented to compare the effect of stress stimuli in PDYN-KO and WT animals, which provides with a numerical estimate to describe the data. Response inhibition was estimated as the percent of the stress-induced effect lost in PDYN KO animals, compared to that observed in WT controls.

Comment: “The results should be more described precisely, and this conclusion comes from which figure.”

Response: The whole paragraph was rephrased to give a more detailed description of the data, including more specific references to the corresponding figures.

Page 13

Comment: “P value??”

Response: Effect sizes and p-values are now shown.

Comment: “Poor description of results, the figure5B and C should described respectively.”

Response: Descriptions of Figures 5B and 5C are now separated and more detailed.

Comment: “Confusing wording”

Response: Sentence rephrased.

Discussion

Page 15

Comment: “Confusing wording”

Response: Sentence rephrased.

Comment: “Description your results correctly”

Response: Sentence slightly reprased.

Comment: “Direct Description”

Response: Kinases are now specified.

Page 16

Comment: “Spelling”

Response: Spelling checked.

Comment: “Confusing wording”

Response: The sentence was simplified.

Comment: “The words "these" and "other" please specify what they refer to”

Response: “These” is now specified.

Comment: “The prior study results should be described briefly to support your conclusion”

Response: Results briefly described.

Comment: “What are the pathways”

Response: Now specified.

Page 17

Comment: “Cited??”

Response: The appropriate reference is now incorporated.

Comment: “Which??”

Response: Now specified.

Comment: “Please do not deny your conclusion immediately after affirmation and state your experimental results precisely”

Response: Sentence rephrased.

Comment: “What do you mean constitutive deletion?”

Response: “Constitutive deletion” was replaced by “mice lacking the PDYN gene”.

Comment: “Do not use this wording”

Response: Sentence rephrased.

Comment: “Wording”

Response: The sentence was deleted because it did not contribute significantly to clarify the argument.

Methods

Page 21

Comment: “Shown in the introduction section”

Response: The sentences were deleted to avoid redundancies.

Page 22

Comment: “PBS or TBS? which one?”

Response: PBS and TBS, depending on the target protein. When the target was a phosphoprotein (e.g. phospho-ERK1/2), TBS was used to avoid spontaneous and unspecific addition of phosphate groups (PBS contains phosphate salts) to the proteins attached to the membranes, which may increase background noise. Otherwise, PBS was used, as it works better under our experimental settings. This issue was further clarified in the text.

Reviewer 2 Report

The authors intend to dissect the molecular mechanisms contributing to the anxiogenic endophenotype in mice lacking the PDYN gene. Overall, the study is interesting, and the results are promising. However, I have the following comments to improve the manuscript:

1.       Authors are suggested to show the alteration of JNK, Akt, mTOR, and ERK expression using proper chemical inhibitors. 

2.       Authors are suggested to show the impact of PDYN-KO in regulating intracellular reactive oxygen species (ROS) levels under stress. 

Author Response

Reviewer #2:

  1. Authors are suggested to show the alteration of JNK, Akt, mTOR, and ERK expression using proper chemical inhibitors.

Response: Enzyme inhibitors are typically used to establish new signaling pathways, or to demonstrate that a particular molecular or physiological event relies on the activation of the target enzyme. While delineating the entire molecular pathways mediating the genetic, morphological and behavioral changes following stressful stimuli is a fundamental question that remains unresolved, the specific goal of the present study was to investigate whether the PDYN gene (and hence dynorphin release) mediates the stress-induced activation of MAPKs, the apoptotic cascade, and/or the Akt/Pea15/mTOR pathway. Thus, for this particular study, using for instance a JNK1/2 inhibitor prior to stress delivery will not help to answer the question of whether or not the PDYN gene mediated JNK stimulation. Please, also note that setting up these kinds of experiments would take months, if not years. Future studies will surely address these interesting questions. Here, we added a comment in the Discussion highlighting the need of using of enzymatic inhibitors to investigate the molecular pathways elicited upon environmental stress stimuli (see page 19).

  1. Authors are suggested to show the impact of PDYN-KO in regulating intracellular reactive oxygen species (ROS) levels under stress.

Response: Certainly, ROS have a critical role in mediating stress responses at a molecular level, and establish an important crosstalk with the opioidergic system. Unfortunately, we no longer have samples left from these animals, and these experiments cannot be carried out for the present study. A comment on this critical issue is provided on page 16.

Reviewer 3 Report

In this study, a comprehensive analysis of the basic neuroadaptations occurring in the cortico-thalamic signaling pathways was carried out. In particular, the apoptotic cascade and Akt were investigated in this work, as well as the role of dynorphins in mediating acute restraint (ARS) and chronic mild (CMS) stress and the changes caused by these molecular cascades in corticothalamic signaling pathways. The authors used a number of molecular biological approaches in their work, in particular, a very detailed IHC analysis was carried out by Western immunoblotting using a whole line of antibodies designed to assess protein expression in the signaling pathways studied by the authors. In the course of a number of experiments, the authors obtained reliable material presented in a statistically correct form. In general, this work deserves high praise, since the results obtained by the authors are correct, and the conclusions made on the basis of the results obtained fully satisfy the correct conditions of the experimental design and also deserve high praise.

During the review of the work, the following minor comments arose:

1. Due to the large abundance of abbreviations and abbreviations, reading the manuscript is difficult. Authors are encouraged to make a correct list of abbreviation.

2. In the Introduction section, the authors generally covered the research problem well, but it is recommended to provide more recent references (along with references 4-6).

3. How was protein loading controlled? What standards were used?

4. In fig. 4H, 5E are lower quality gels than the other figures. What is it connected with?

 After elimination of these minor comments, the work can be published in the IJMS.

Author Response

Reviewer #3:

  1. Due to the large abundance of abbreviations and abbreviations, reading the manuscript is difficult. Authors are encouraged to make a correct list of abbreviation.

Response: An abbreviation list was included in the current version of the manuscript.

  1. In the Introduction section, the authors generally covered the research problem well, but it is recommended to provide more recent references (along with references 4-6).

Response: Some more recent references supporting the arguments were added.

  1. How was protein loading controlled? What standards were used?

Response: Loading controls for phospho-activated kinases were the respective total forms of the kinases. For example, phospho-Thr183/Tyr185 JNK1/2 immunodensity was normalized by the corresponding total-JNK1/2 signal in the same gel. Non-phospho-activated proteins (FADD, PARP, etcetera) were controlled either by beta-actin, when comparing basal levels in WT versus PDYN-KO mice, or by alpha-tubulin when comparing stressed versus non-stressed animals. Please, note that stress was reported to alter beta-actin levels, and such housekeeping protein should not be used in experiments involving environmental stress (reference cited in the manuscript). All figures show the corresponding immunoblot of the selected housekeeping protein, and a whole paragraph in the Materials and Methods section is dedicated to this matter (see page 22). The next paragraph (page 23) explains how the normalization procedure (i.e. correction by the loading control protein) was performed, as well as the calculations to compare the immunodensities across the gels. As per the second concern, we added a new sentence in the methodological section indicating molecular weight standards used (see first paragraph in page 22).

  1. In fig. 4H, 5E are lower quality gels than the other figures. What is it connected with?

Response: Immunoblots in Figures 4H (phospho-/total-mTOR) and 5E (phospho-/total-Pea15) have been swapped. While the current mTOR selected blot is objectively of better quality, the immunoblots for Pea15 improved just slightly. Please, note that WB is a ‘crafty’ technique, and some experimental gels simply run and transfer better than others. This issue is especially concerning when the target protein has a very high or very low molecular size, as they run to the top or bottom edges of the gel, respectively. Gel edges are typically more sensitive to temperature and humidity variations, and may be more vulnerable to experimenter’s handling. All these factors may result in the typical ‘smiling’ or ‘waving’ immunoblotting patterns. Both mTOR (250 kDa) and Pea15 (15 kDa) belong to these types of proteins in the high/low end of molecular sizes. Therefore, the chances of poorer quality in mTOR and Pea15 immunoblots are greater than those of other protein targets. In any case, we do an honest selection of those gel portions containing a combination of samples representing the mean value of each experimental group. Sometimes, those combinations are located in gels that just did not run as smooth as wanted, which might be the case of these experiments. Despite their poorer quality, we believe that current and former immunoblot portions selected for figures 4H and 5E indeed represent the overall data and may have sufficient quality for publication.

Round 2

Reviewer 1 Report

Comments to AUTHORS:

This article has been revised in response to the suggestions, and deficiencies in the experiments have been explained in the discussion section.